



# Assessment of Upscaling Methodologies for Daily Crop Transpiration using Sap-Flows and Two-Source Energy Balance Models in Almonds under Different Water Status and Production Systems

Manuel Quintanilla-Albornoz[1], Xavier Miarnau[2], Ana Pelechá[1], Héctor Nieto[3] and Joaquim Bellvert[1]

[1]Efficient Use of Water in Agriculture Program, Institute of Agrifood Research and Technology, Fruitcentre, Parc AgroBiotech, Lleida, 25003, Spain.
[2]Production Program, Institute of Agrifood Research and Technology, Fruitcentre, Parc AgroBiotech, Lleida, 25003, Spain
[3] Institute of Agricultural Sciences, ICA-CSIC, Madrid, 28006, Spain.

*Correspondence to*: Manuel Quintanilla-Albornoz (manuel.quintanilla@irta.cat)

**Abstract.** The daily transpiration ($T_d$) is crucial for both irrigation water management and increasing crop water productivity. The use of the remote sensing-based two-source energy balance model (TSEB) has proven to be robust in estimating plant transpiration and evaporation separately for various crops. However, remote sensing models provide instantaneous estimations, so daily upscaling approaches are needed to estimate daily fluxes. Daily upscaling methodologies have not yet been examined to upscale solely transpiration in woody crops. In this regard, this study aims to evaluate the proper image acquisition time

throughout the day and four methodologies to retrieve $T_d$ in almond trees with different production systems and water status. Hourly transpiration ($T_h$) was estimated using the TSEB contextual approach ($T_h$-TSEB) with high-resolution imagery five times during two diurnal courses. The tested methodologies were the following: the simulated evaporative fraction variable ($EF_{sim}$), irradiance ($Rs$), reference evapotranspiration (ETo) and potential evapotranspiration (ETp). These approaches were firstly evaluated with in situ sap flow (T-SF) data and then applied to the $T_h$-TSEB. Daily T-SF showed significant differences

among production systems and levels of water stress. The $EF_{sim}$ and ETp methods correlated better with measured T-SF, and reduced the underestimation observed using the $Rs$ and ETo methods, especially at noon in the severely water stressed trees. However, the daily upscaling approaches applied in the TSEB ($T_d$-TSEB) failed to detect differences between production systems. The lack of sensibility of $T_d$-TSEB among production systems poses a challenge when estimating $T_d$ in canopies with discontinuous architectural structures. The improvement of ETp estimations or more sophisticated ETp models could solve

this issue.

## 1 Introduction

Almond is one of the high-value crops with the greatest water usage (Goldhamer and Fereres 2017; López-López et al. 2018). In Spain, a paradigm change is taking place with the introduction of new intensified almond production systems with more planar designs (Iglesias and Echeverria 2022), which may complicate the accurate estimation of evapotranspiration (ET) using

remote sensing models. Thus, since the expansion of almond production is occurring in a context of increasing water scarcity,





many studies have focused on quantifying its water usage in different environments and water regimes. From a water management point of view, there is particular interest in validating the daily ET and its components, transpiration (T) and evaporation (E), in this crop and under different production systems and water status. This is relevant because almond is considered a drought-tolerant species able to control water loss through stomatal closure, which has been identified as a

common and early event in plant response to water deficit (Castel and Fereres, 1981, Escalona et al. 1999, Chaves et al. 2002). Romero et al. (2006) also showed that the influence of the evaporative demand of the atmosphere on stomatal behavior was higher under well-watered compared to water-stressed almonds. The same study also demonstrated that water-stressed almonds restricted stomatal activity earlier in the morning when atmospheric vapor pressure deficit (VPD) was still low. As a result, maximum T values occurred during this period and were significantly higher than those observed in well-watered almonds.

Accurate in-field quantification of crop ET and the partition components E from soil and plant T is very useful for both irrigation water management and increasing crop water productivity (Zhang et al. 2021). Consequently, several methodologies have been developed to address this objective (Evett and Tolk 2009). Of these, remote sensing thermal-based surface energy balance models have shown their utility in retrieving ET in a wide range of environments and ecosystems (Shuttleworth and Wallace 1985; Bastiaanssen et al. 1998; Drexler et al. 2004; Overgaard et al. 2006; Allen et al. 2007; Timmermans et al. 2007;

Kalma et al. 2008; Kustas and Anderson 2009). The advantage of using remote sensing lies in the possibility of monitoring heterogeneous surfaces over a wide range of spatial resolutions and thereby generating operational ET products (Kalma et al. 2008). One such model that calculates T and E explicitly is the two-source energy balance (TSEB), which was initially developed by Norman et al. (1995) and Kustas and Norman (1999). The TSEB approach has demonstrated its robustness in accurately estimating plant ET, T, and E across diverse surface conditions and a wide range of landscapes (Kustas and

Anderson 2009; Kustas et al. 2019; Gómez-Candón et al. 2021; Gao et al. 2023; Knipper et al. 2023). Additionally, the separate T and E outputs provide the advantage of simultaneously evaluating canopy stress and directly quantifying plant water consumption. This information can be valuable for enhancing water use efficiency in agricultural and environmental management. Moreover, T is also linked to plant productivity as the exchange of both water and carbon between the atmosphere and the plant is conveyed via the leaf stoma.

To estimate T, the use of very high resolution thermal and multispectral imagery allows for the direct estimation of canopy ($T_c$) and soil temperature ($T_s$), facilitating the retrieval of ET partitioning, through use, for example, of the TSEB contextual approach (TSEB-2T) model (Nieto et al. 2019; Nassar et al. 2020; Gao et al. 2023). Models for estimating ET fluxes based on remote sensing, however, can only be used to derive an instantaneous ET at the time of clear-sky satellite or aircraft overpass. Thus, the selection of a proper overpass time and the development of upscaling algorithms to extrapolate ET from instantaneous

to daily scale are of special interest for the management of crop water consumption. Current thermal infrared (TIR) polar orbiting satellites, such as Landsat, Sentinel-3 or the moderate-resolution imaging spectroradiometer (MODIS) on board Terra, have an overpass time close to 10:00 am (mean locator solar time). However, several studies suggest that the best accuracies in ET retrievals would be captured better in the early afternoon (Delogu et al. 2012; Anderson et al. 2021). Bellvert et al. (2014) also showed that early afternoon was the most appropriate moment to detect maximum differences in Tc between well-





watered and water-stressed crops. For this reason, in coming years new TIR satellite missions including TRISHNA (Thermal infraRed Imaging Satellite for High Resolution Natural resource Assessment) (Lagouarde et al. 2018), SBG (Surface Biology and Geology) (Basilio et al. 2022), or LSTM (Land Surface Temperature Monitoring) (Koetz et al. 2018) are planned at an overpass time around 13:00 hours (GMT time).

Daily upscaling of ET fluxes is commonly performed by assuming a constant relationship over the course of the day between instantaneous ET and a reference meteorological forcing that can be computed at hourly and daily timesteps (Crago and Brutsaert 1996; Cammalleri et al. 2014). This hypothesis is generally known as self-preservation (Crago and Brutsaert 1996). Generally, the most commonly used methods for upscaling ET are: the evaporative fraction (EF) method, the solar radiation (Rs) method, the stress factor method and the canopy resistance method (Hoedjes et al. 2008; Delogu et al. 2012; Cammalleri et al. 2014; Jiang et al. 2021; Nassar et al. 2021). Experimental studies have shown that the EF method, which is based on the ratio between latent heat flux (LE) and the available energy at the surface (AE), is relatively stable during midday hours for days with clear sky conditions, but significantly higher during early morning and late afternoon. These differences in EF during the day cause a systematic underestimation of daytime average values under wet conditions (Shuttleworth et al. 1989; Brutsaert 1992; Crago and Brutsaert 1996; Lhomme and Elguero 1999; Gentine et al. 2007). To address this challenge, Hoedjes et al. (2008) introduced a parameterization of the diurnal EF pattern based on the primary atmospheric forcing parameters: Rs and relative humidity (RH). Implementing this approach, known as EFsim, Delogu et al. (2012) successfully reduced the overestimation associated with the EF method from 15.8% to 6.5%.

Additionally, while estimating the instantaneous AE at a specific time can be relatively straightforward using thermal imagery and meteorological data, determining daily AE needs daily course measurements or estimates of net radiation (Rn) and soil heat flux (G), which can be challenging. Given that the diurnal pattern of AE is primarily influenced by Rs, it has become a common practice to use Rs as a reference variable for the estimation of daily ET fluxes from instantaneous measurements (Jackson et al. 1983; Zhang and Lemeur 1995). The use of Rs in the context of remote sensing applications has fewer requirements than the EF method, with the latter needing auxiliary information such as Rn that can be complex to measure and may further limit operational utility. When used the Rs upscaling method, both Cammalleri (2014) and Nassar (2021) improved daily ET compared to EF methods.

Another upscaling method that has been proposed is the stress factor method. This approach employs the reference evapotranspiration (ETo) or potential evapotranspiration (ETp) as a reference variable, which inherently accounts for the key meteorological factors influencing the evaporative process (Trezza 2002; Delogu et al. 2012). Trezza (2002) found a constant ratio between ET and ETo during the daytime and employed it to estimate daily ET using remote sensing estimations, achieving better results compared to EF upscaling methods. However, Cammalleri (2014) obtained similar results when using both the EF method and the ETo to estimate daily ET in sites without stress conditions. For their part, Delogu et al. (2012) evaluated the use of ETp as a reference variable and obtained worse results compared to the EF method for a dataset with stress events. This was attributed to the fact that the AE followed both stressed and unstressed ET patterns, whereas ETp often increased independently of the water stress levels.



Nevertheless, the aforementioned upscaling methods reported in the literature in agricultural ecosystems have only been
validated against daily ET, usually over sites with eddy-covariance flux towers, with a footprint with mixed information on
the spatial variability (Cammalleri et al. 2014, Xu et al. 2018, Jiang et al. 2021). Therefore, to the best of our knowledge, the
use of upscaling methodologies to estimate daily T (Td) based on instantaneous T values has not been previously examined.
Furthermore, the diurnal pattern of T has a different response between well-watered and water-stressed crops (Poni et al. 2009,
Tuzet et al. 2003), and this different response would also depend on the stomatal control of each species to soil water and vapor
pressure deficits. Thus, the hypothesis of this study is that upscaling methods may have different responses for water-stressed
and well-watered almond trees (Sánchez et al. 2021; Ĉekalović et al. 2022; Iglesias and Echeverria 2022; Peddinti and Kisekka
2022; Knipper et al. 2023). Therefore, the main purpose of this study is to evaluate different $T_d$ upscaling methodologies in
almond trees under different production systems and water regimes using sap flow measurements. This study aims to contribute
to our understanding and establish a reference for upscaling remote sensing canopy T in woody crops, which is crucial in
mapping daily ET partitioning from field to global scales.

## 2 Materials and Methods

### 2.1 Trial location and design

This study was conducted in an almond orchard situated at the experimental station of the IRTA (Institute of Agrifood Research
and Technology) in Les Borges Blanques, Spain (41°30'31.89''N; 0°51'10.70''E, 323 m elevation) (Fig. 1a). The almond
orchard was planted in June 2009, with "Marinada" used as the scion cultivar onto an INRA GF-677 rootstock. Three almond
production systems were evaluated: open vase with minimal pruning (MP) spaced at 5.5 x 3.5 m, central axis at 5 x 3 m, and
hedgerow at 4.5 x 3 m (Fig. 1b). The orchard was situated on a clay loam-textured soil, with a depth ranging from 1.6 to 2 m.
The study site has a Mediterranean climate, with an average annual rainfall of 364 mm and an average annual
evapotranspiration of 1088 mm. Two different dates were selected to assess the diurnal course of T: 29th June and 19th August
2022. Figure 2 displayed the meteorological conditions during the campaign.





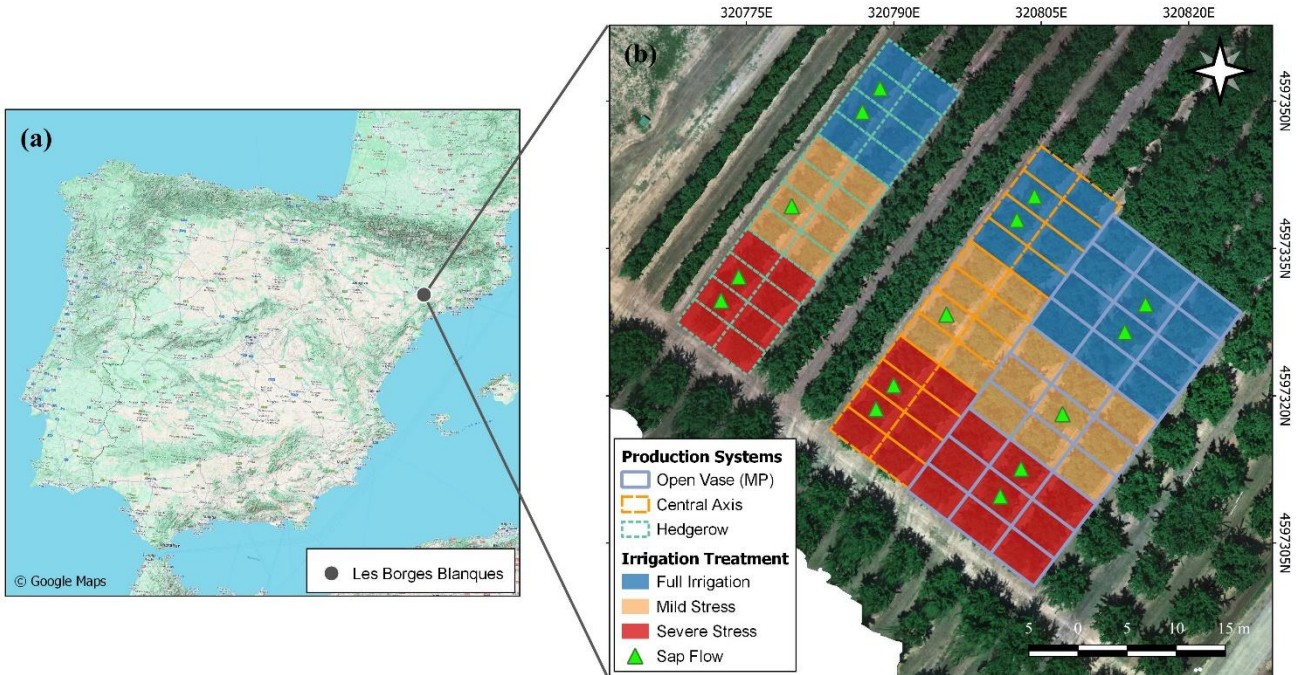

**Figure 1: Location of the almond orchard in Les Borges Blanques (a) and experimental design of the orchard, showing in different colors the three production systems and the three irrigation treatments (b).**


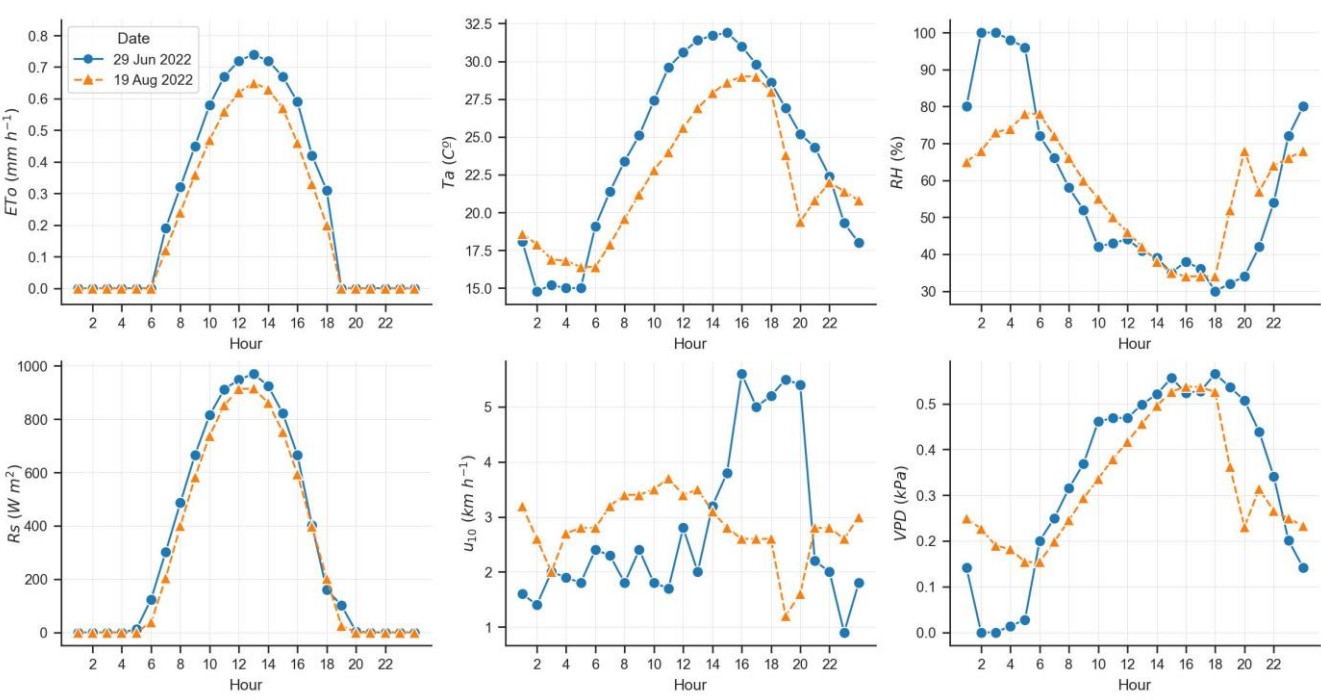

**Figure 2. Meteorological conditions at hour scale during the flight campaign.**





The orchard was irrigated using a drip irrigation system. In the open vase (MP) system, two laterals were positioned on each
side of the tree at 40 cm, with a dripper placed every 70 cm and a water discharge rate of 2.2 l h$^{-1}$. The central axis and
hedgerow systems had a lateral pipe along the row line, with drippers placed at 60 cm intervals with a water discharge rate of
3.8 l h$^{-1}$ per dripper. Daily irrigation was scheduled on a weekly basis to complement potential crop evapotranspiration (ETc)
using: ETc = (ETo x Kc) – effective rainfall, as described by Allen et al. (1998). ETo was obtained from a meteorological
station within Catalonia's official network of meteorological stations (SMC, https://ruralcat.gencat.cat/web/guest/agrometeo),
situated 500 m away from the study site. The ETo is estimated using the FAO56 Penman-Monteith method (Allen et al. 1998).
Kc refers to the crop coefficient. The Kc was assigned based on different phenological stages, following Goldhamer and Girona
(2012). The assigned Kc values were: $Kc_1$ = 0.70 (April), $Kc_2$ = 0.95 (May), $Kc_3$ =1.09 (June), $Kc_4$ = 1.15 (July), $Kc_5$ = 1.17
(August), and $Kc_6$ = 1.12 (September). Effective rainfall was determined following the method outlined by Olivo et al. (2009),
which considers half of the precipitation for days with a single event exceeding 10 mm, otherwise, it is considered zero. Three
irrigation treatments were implemented for each production system during the 2021 and 2022 growing season: (i) Full
irrigation, where irrigation matches ET requirements (100% Etc); (ii) mild stress, irrigated at 50% Etc; and (iii) severe stress,
irrigated at 20% Etc. The water applied was quantified using digital water meters (CZ2000-3M, Contazara, Zaragoza, Spain).

## 2.2 Sap flow measurement

Sap flow sensors offer substantial advantages, enabling the continuous and automated measurement of sap movement for each
plant with a high temporal resolution (Smith and Allen 1996; Forster 2017; Fernandez 2001). When properly calibrated, these
sensors can measure the T for the entire plant (López-Bernal et al. 2010; Forster 2017; Noun et al. 2022). Among the sap flow
measuring methods available, the compensation heat pulse (CHP) has been suggested as a tool for detecting water stress and
for irrigation scheduling purposes (Fernandez 2001; Alarcón et al. 2005). Therefore, the CHP sap flow method combined with
the calibrated average gradient technique was employed to estimate the T. The sap flow system consists of a 2 mm diameter
4.8 W stainless steel heater and two temperature sensors positioned 10 and 5 mm downstream and upstream of the heater,
respectively. Each temperature sensor is embedded with two E-type thermocouples (chrome-constantan wire) spaced 10 mm
apart along the needle. The heat pulse velocity at 5 and 15 mm below the cambium is used to calculate the sap flow density
across the trunk radius. The sap flow system was developed by the IAS-CSIC laboratory. For further specifications, refer to
Villalobos et al. (2009). Sap flow data were collected every 15 minutes and stored in a CR1000 datalogger (Campbell Scientifc
Inc., Logan, UT, USA).

Sap flow sensors were installed in each production system, monitoring two trees from the full irrigation and severe stress
treatments, and one tree from the mild stress treatments, as shown in Figure 1b. They were installed at 0.5 m above the ground.
Each sap flow transpiration (T-SF) underwent correction for wound and azimuthal effects (López-Bernal et al. 2010) using
actual T measured by a water balance method ($T_{wb}$) on July 13, 2022. The $T_{wb}$ was calculated using Eq. (1).





$$T_{wb} = P + I_R - \Delta SWC - DP - \text{E}, \qquad (1)$$

Where $P$ is precipitation, $I_R$ is the amount of water applied through irrigation, $\Delta SWC$ is the difference in soil water content (SWC) between two consecutive days, $DP$ is deep percolation and E corresponds to evaporation. $P$, $DP$ and $I_R$ were considered zero because the water balance was calculated for days without $P$ and $I_R$ applied. Additionally, the soil was covered with plastic sheeting during these days to prevent E fluxes (E ≈ 0). Differences between $T_{wb}$ and T-SF measurements were assumed to remain constant throughout the season, as demonstrated by Espadafor et al. (2015). The calibrated T-SF was used to calculate both the accumulated hourly T ($T_h$-SF) and the accumulated daily T ($T_d$-SF).

The SWC was measured using a neutron probe at intervals of 20 cm down to a depth of 180 cm (Campbell Pacific Nuclear Scientific, Model 503). The tubes used for SWC measurements were installed to cover one quarter of the planting area. In each tree, two groups of three tubes were installed in parallel, positioned below the emitter, at a quarter of the inter-row distance, and at half of the inter-row distance. Soil sampled were taken at the time of tubes installation to estimate the volumetric moisture content ($cm^3$ of water $cm^{-3}$ of soil). This measurement was then used to calibrate the neutron probe readings.

### 2.3 Field measurement

#### 2.3.1 Stem water potential, stomatal conductance, leaf transpiration and leaf area index

Stem water potential ($\Psi_s$), stomatal conductance ($g_s$) and leaf transpiration ($E_{leaf}$) were measured at 7:00, 9:00, 12:00, 14:00 and 16:00 solar time during the UAV flight campaign and in the same trees where sap flow sensors were installed. The measurement of $\Psi_s$ followed the protocol outlined by McCutchan and Shackel (1992). The $\Psi_s$ was determined by measuring three shaded leaves from each tree. Prior measurement, each leaf was enclosed in a plastic bar covered with aluminium foil for one hour to equalize the water potential between the leaf, stem, and branches. A pressure chamber (Plant Water Status Console, Model 3500; Soil Moisture Equipment Corp., Santa Barbara, CA) was utilized to obtain the $\Psi_s$ in all measurement within one hour. The $g_s$ and $E_{leaf}$ were measured using the LI-600 porometer/fluorometer (LI-COR Inc., Lincoln, NE, USA). Three sunny leaves were measured in each tree concomitant to image acquisition. The leaf area index (LAI) was determined for trees equipped with sap flow sensors using the LAI-2200 Plant Canopy Analyzer (PCA) (LI-COR Inc., Lincoln, NE, USA) The LAI was measured in each flight date around midday. The LAI measurement procedure involved one measurement taken above the tree and four below the tree. The incident radiation above the tree was recorded in an open area using five sensor rings. A single measurement was taken in each cardinal direction (N, S, E and W) beneath the tree. The LAI was subsequently estimated from the vertical profile of the crown using the FV2200 v. 2.1.1 software. The accuracy of LAI estimations was 0.57 $m^2$ $m^{-2}$ (Quintanilla-Albornoz et al. 2023).

### 2.3.2 Image acquisition campaign

Ten flights were conducted on June 29 and August 29 of 2022 with UAV Dronehexa XL (DRONETOOLS, Seville, Spain). On each day, five flights were conducted at 7:00, 9:00, 12:00, 14:00 and 16:00 GMT. The UAV was outfitted with a Micasense





RedEdge-MX multispectral camera (Micasense, Northlake Way, Seattle, USA) and a FLIR SC655 thermal camera (FLIR Systems, Wilsonville, OR, United States). Micasense RedEdge-MX captures images in five spectral bands at wavelengths of 475 ± 20 nm, 560 ± 20 nm, 668 ± 10 nm, 717 ± 10nm, and 840 ± 40 nm. FLIR SC655 has a spectral response in the range of 7.5–13 µ. The flights were carried out at a height of 50 m above ground level to capture multispectral and thermal images with

spatial resolutions of 0.03 m and 0.06 m, respectively.

All images were subjected to radiometric, atmospheric and geometric correction. The FieldSpec 4 Standard-Res spectroradiometer (Malvern Panalytical, Inc., United Kingdom) was used to acquire in situ spectral measurements on various ground target simultaneously with the image acquisition for radiometric calibration. The FieldSpec 4 Standard-Res spectroradiometer has an optical resolution of 3-10 mm and a wavelength response between 350 and 2500 nm. Before

conducting spectral measurements on the ground targets, the spectroradiometer was calibrated using white reference panel (white color SpectralonTM) and a dark reference. The thermal sensor underwent radiometric calibration in the laboratory using a blackbody (model P80P, Land Instruments, Dronfield, United Kingdom). Additionally, in-situ temperature measurements were acquired using an SI-111-SS Apogee infrared radiometer connected to an Apogee AT-100 microCache Bluetooth micrologger (Apogee instruments Inc, Logan, UT, USA). The mosaicking process, as well as the generation of the digital

elevation model (DEM) and the digital surface model (DSM), were performed using Agisoft Metashape Professional software (Agisoft LLC., St. Petersburg, Russia). Geometric and radiometric corrections was conducted using QGIS 3.4 (QGIS 3.4.15).

**2.4 TSEB model description**

The TSEB scheme, initially introduced by Norman et al. (1995) and further refined by Kustas and Anderson (2009), was utilized to estimate T employing high-resolution images. The TSEB is an energy balance models that assumes net surface

radiation ($R_n$) is primarily distributed among sensible heat flux ($H$), latent heat flux ($LE$) and soil heat flux ($G$). Consequently, the $LE$ (W m$^{-2}$) is calculated as the residual of the surface energy equation by Ep. (2.1), Eq. (2.2) and Eq. (2.3):

$$LE \approx R_n - H - G, \tag{2.1}$$

$$LE_s \approx R_{n,s} - H_s - G, \tag{2.2}$$

$$LE_c \approx R_{n,c} - H_s, \tag{2.3}$$

Where the subscripts $_C$ and $_S$ refer to the energy fluxes of the canopy and soil, respectively. The Campbell and Norman (1998) canopy transfer model, considering a rectangular clumping index, was employed to estimate $R_{n,s}$ and $R_{n,c}$, as described by Parry et al. (2019) and Quintanilla-Albornoz et al. (2023). $G$ was assumed as a constant fraction of $R_{n,s}$ of around 0.35. A series resistance scheme was utilized, dividing $H$ into soil ($H_s$) and canopy ($H_c$) as shown in Eq. (3.1), Eq. (3.2) and Eq. (3.3):

$$H_s = \rho C_p \frac{T_s - T_{ac}}{r_s}, \tag{3.1}$$

$$H_c = \rho C_p \frac{T_c - T_{ac}}{r_x}, \tag{3.2}$$



$$H_s + H_c = \rho C_p \frac{T_{ac} - T_a}{r_a},$$ (3.3)

where ρ is the air density, $C_p$ is the specific heat of air, $T_s$ is the soil temperature, $T_c$ is the canopy temperature, $T_a$ is the air temperature, $T_{ac}$ is the temperature in the canopy air space, equivalent to the aerodynamic temperature, $r_s$ is the resistance to heat flow in the boundary layer immediately above the soil surface, $r_x$ is the total boundary layer resistance of the complete

canopy leaves, and $r_a$ is the aerodynamic resistance to turbulent heat transport between the air canopy layer and the overlying air layer. The resistances were derived according to Kustas and Norman (1999) and Norman et al. (1995).

The contextual approach of the TSEB model (TSEB-2T) was evaluated in this study and is available online at https://zenodo.org/doi/10.5281/zenodo.594732. The TSEB-2T was applied with direct measurements of $T_c$ and $T_s$ from high-resolution thermal images. $T_c$ and $T_s$ were obtained with a supervised image classification based on using the DSM and the

soil-adjusted vegetation index (SAVI). SAVI was chosen due to its ability to reduce the impact of ground brightness in the near and shortwave infrared wavelengths, which enhances the contrast between vegetation and the ground surface (Qi et al. 1994). Pixel were classified as canopy if they exhibited a DSM greater than 1.5 m and a SAVI greater than 0.2. Pixels that did not meet these conditions were classified as pure soil. These layers were employed to retrieval the $T_c$ and $T_s$ from thermal images. Finally, the hourly T in mm ($T_h$-TSEB) was estimated using: $1000 \times 3600 \times LE_c / (\rho_w \lambda)$, where $\rho_w$ is the density of

water (assumed to be 1,000 kg m⁻³) and λ is the latent heat of vaporization (J kg⁻¹): $\lambda = 1e^6 \times (2.501 - 0.002361\, T_a)$. All biophysical traits required for TSEB models, the fractional canopy cover (*fc*), canopy height (*hc*) and canopy width (*wc*), were obtained using the multispectral and DSM high resolution images. For additional details on the biophysical traits' procedure, refer to Quintanilla-Albornoz et al. 2023.

**2.5 Models evaluated to upscale daily transpiration**

The self-conservation method is the most commonly used approach to upscale ET fluxes from instantaneous measurements. This assumes a constant relationship between the instantaneous ET and some meteorological variables over time under certain conditions. According to Cammalleri (2014), the relationship between instantaneous measurement of ET fluxes and a reference variable can be illustrated using Eq. (4):

$$ET_d = \beta \frac{1}{\lambda} \frac{\lambda LE_t}{X_t} X_d,$$ (4)

where $\lambda LE_t$ is the instantaneous latent heat flux at the acquisition time *t*, $X_t$ and $X_d$ are the values of the reference variable at the acquisition time *t* and during the day *d*, and *β* represents a correction factor to account for potential biases or nighttime ET. This paper evaluates four self-preservation approaches, elucidated below, along with their implications for estimating $T_d$ in almond crops.



### 2.5.1 Simulated evaporative fraction variable (EFsim) method

The $EF_{sim}$ is based on the evaporative fraction (EF) method. The EF method assumes that the ratio between *LE* and AE is relatively constant during the day. Following Eq. (5.1), Eq. (5.2) and Eq. (5.3), we can obtain the daily *LE* fluxes:

$$EF = \frac{LE}{AE} \tag{5.1}$$

$$AE = R_n - G \tag{5.2}$$

$$LE_d = AE_d \times EF \tag{5.3}$$

where $LE_d$ and $AE_d$ correspond to daily accumulated *LE* and AE, respectively. *Rn* can be determined from remote sensing data using Eq. (6):

$$R_n = (1 - \alpha) \cdot R_S + \varepsilon \cdot R_{atm} - \varepsilon \cdot \sigma \cdot T_{rad} \tag{6}$$

where $\alpha$ corresponds to the albedo, ε the surface emissivity, $R_{atm}$ the atmospheric longwave radiation, σ the Stefan-Boltzman constant, and $T_{rad}$ the radiometric temperature. To avoid daily measurement of $R_n$ and *G*, the AE can be extrapolated from

instantaneous AE estimated through thermal imagery and *Rs*, following the methods proposed by Jackson et al. (1983) and Deluge et al. (2012), as expressed in Eq. 7:

$$AE_d = Rs_d \frac{AE_t}{Rs_t} \tag{7}$$

where $AE_t$ represents the instantaneous AE estimated through thermal imagery, $Rs_d$ the daily *Rs*, and $Rs_t$ is the *Rs* at the measurement time. According to Hoedjes et al. (2008), the daily pattern of EF can be simulated as a function of *Rs* and RH,

as in Eq. (8.1). However, $EF_{sim}$ is a theoretical curve and must be adjusted using real EF values with Eq. (8.2):

$$EF_{sim} = 1.2 - \left(0.4 \frac{Rs}{1000} + 0.5 \frac{RH}{100}\right) \tag{8.1}$$

$$EF_{adj} = EF_{sim} \frac{EF_{t,obs}}{EF_{t,sim}} \tag{8.2}$$

where *Rs* is in W m$^{-2}$ and RH is in percentage. Additionally, $EF_{t,obs}$ represents actual EF values estimated using remote sensing imagery based on Eq. (5.1), and $EF_{t,sim}$ is the $EF_{sim}$ at the time of $EF_{t,obs}$. Finally, the $EF_{sim}$ method employs Eq. (5.3) with an

EF estimated using Eq. (8.2) and $AE_d$ estimated using Eq. (7) to estimate $LE_d$.

### 2.5.2 Incoming shortwave solar radiation (*Rs*) approach

An alternative strategy consists of replacing AE as a reference variable with the *Rs*. This method is founded on the principle that *Rs* is the primary radiation flux during the day, resulting in a strong correlation and associated variations between actual ET and *Rs* (Jackson et al. 1983; Delogu et al. 2012, Nassar et al. 2021). Thus, $LE_d$ can be estimated with Eq. (9):

$$LE_d = Rs_d \frac{LE_t}{Rs_t} \tag{9}$$

where $Rs_d$ corresponds to daily *Rs* and $Rs_t$ is the *Rs* at the time that LE was estimated.





### 2.5.2 Stress factor approach

The stress factor approach involves upscaling the instantaneous ET using either reference (ETo) or potential evapotranspiration (ETp), as depicted in Eq. (10):

$$ET_d = SF \cdot ET_O(ET_p) \tag{10}$$

The stress factor is defined as the ratio between ET and instantaneous ETo or ETp (*SF = ET/ETo (or ETp)*). The ETo was obtained using the FAO-56 method (Allen et al., 1998). ETp was estimated using the Penman Monteith one-source energy balance model, and forcing it with meteorological data and the actual LAI (Allen et al., 1998). The ETp obtained from the Penman Monteith model is available in the Python programming language at https://zenodo.org/doi/10.5281/zenodo.594732.

Meteorological data were obtained from the weather station of the Meteorological Service of Catalonia located near the experimental orchard.

The $EF_{sim}$, *Rs*, ETo and ETp upscaling methods were used to estimate $T_d$ from $T_h$-SF measurements and from $T_h$-TSEB estimations. The $T_d$ obtained using the $EF_{sim}$, *Rs*, ETo and ETp upscaling methods from $T_h$-SF measurements was called $T_d$-SF-$ET_{sim}$, $T_d$-SF-*Rs*, $T_d$-SF-ETo and $T_d$-SF-ETp, while the $T_d$ estimated from $T_h$-TSEB estimations was called $T_d$-TSEB-$ET_{sim}$,

$T_d$-TSEB-*Rs*, $T_d$-TSEB-ETo and $T_d$-TSEB-ETp, respectively.

## 3 Results

### 3.1 Biophysical traits and physiological measurements

Table 1 shows an analysis of variance (ANOVA) of the main biophysical traits and Table 2 the average of each biophysical variable for each production system and irrigation treatment. The fractional canopy cover (*fc*) significantly varied between

production systems, with open vase (MP) and hedgerows presenting the highest and lowest values, respectively. The average *fc* for each production system was 0.56, 0.50 and 0.47 for open vase (MP), central axis and hedgerow, respectively. Canopy height (*hc*) significantly varied between production systems, irrigation treatments and their interaction. Overall, taller trees were observed in the open vase (MP) system. However, open vase (MP) and hedgerow systems led to smaller trees in the severe stress treatment, whereas the central axis had the smallest trees in the mild stress treatment. The measured LAI did not

show significant differences among production systems or irrigation treatments.

| Variable | Date | PS | TRT | PSxDate | TRTxDate | PSxTRT | PSxTRTxDate |
|---|---|---|---|---|---|---|---|
| $f_c$ | ns | 0.008 | ns | ns | ns | ns | ns |
| $h_c$ | 0.0004 | <.0001 | <.0001 | ns | ns | <.0001 | ns |
| LAI | ns | ns | ns | ns | ns | ns | ns |
| $\Psi_s$ | ns | 0.0059 | <.0001 | ns | 0.0044 | 0.0398 | ns |
| $g_s$ | ns | ns | <.0001 | ns | 0.0046 | 0.0152 | ns |
| $E_{leaf}$ | 0.0003 | 0.0098 | <.0001 | ns | 0.0321 | 0.0188 | ns |
| $T_d$-SF | 0.0001 | 0.0033 | <.0001 | ns | ns | 0.0111 | ns |
| $T_h$-$SF_{morning}$ | 0.0003 | <.0001 | <.0001 | ns | ns | 0.015 | ns |





| Variable | Date | PS | TRT | PSxDate | TRTxDate | PSxTRT | PSxTRTxDate |
|---|---|---|---|---|---|---|---|
| $T_h$-$SF_{midday}$ | ns | ns | <.0001 | ns | ns | ns | ns |
| $T_h$-$SF_{afternoon}$ | ns | 0.005 | <.0001 | ns | 0.011 | 0.001 | ns |

**Table 1. Analysis of variance (three-way ANOVA) testing the effect of date, production system (PS) and irrigation treatment (TRT) on biophysical traits, physiological parameters, and hourly ($T_h$-SF) and daily transpiration ($T_d$-SF) measured by sap flow sensors. P values less than 0.05 were considered statistically significant.**


| Production system | Irrigation treatment | $fc$ | $fc$ | LAI | $T_d$-SF |
|---|---|---|---|---|---|
| Open Vase | Full irrigation | 0.61 a | 5.82 a | 3.12 | 4.61 a |
| | Mild stress | 0.57 a | 5.42 b | 2.8 | 3.8 ab |
| | Severe stress | 0.51 a | 5.01 c | 2.96 | 1.3 c |
| Central Axis | Full irrigation | 0.53 ab | 4.11 d | 3.08 | 3.75 b |
| | Mild stress | 0.5 ab | 4.07 d | 3.27 | 2.6 b |
| | Severe stress | 0.48 ab | 3.5 e | 3.16 | 1.54 c |
| Hedgerow | Full irrigation | 0.44 b | 4.02 d | 2.61 | 3.37 b |
| | Mild stress | 0.5 b | 4.78 c | 3.7 | 3.59 b |
| | Severe stress | 0.49 b | 4.18 d | 3.65 | 0.99 c |

**Table 2. Comparison of the main biophysical traits measured during the flight campaign. Different letters mean significant differences at p < 0.05 using Tukey's honest significant difference test considering the interaction between production system and irrigation treatment.**

**3.1.1 Stem water potential, stomatal conductance and leaf transpiration**

The diurnal patterns of $\Psi_s$, $g_s$, and $E_{leaf}$ exhibited variations primarily attributed to the irrigation treatment (Fig. 3). These variations led to significant differences in tree daily average $\Psi_s$, $g_s$, and $E_{leaf}$ among the different irrigation treatments (Table 1). Moreover, the interaction between production system and irrigation treatment (PSxTRT) had a significant impact, primarily attributable to the central axis subjected to the mild stress treatment. The central axis under the mild stress treatment exhibited

values comparable to those observed in the severe stress treatment. The daily pattern of $\Psi_s$ exhibited significant differences between irrigation treatments as early as 7:00 hours. In contrast, discernible significant differences between irrigation treatments for $g_s$ and $E_{leaf}$ were evident as early as 9:00 hours. Differences in $\Psi_s$, $g_s$ and $E_{leaf}$ between irrigation treatments remained evident until 16:00 hours. The peak disparities in $\Psi_s$, $g_s$ and $E_{leaf}$ among irrigation treatments were observed around 12:00 hours. During this time, $\Psi_s$ had its most reduced values with an average of -1.35 MPa in the full irrigation, -1.86 MPa

in the mild stress and -2.30 MPa in the severe stress treatments. Simultaneously, $g_s$ attained its maximum values with an average of 0.41 mmol m$^{-2}$ s$^{-1}$, 0.25 mmol m$^{-2}$ s$^{-1}$, and 0.12 mmol m$^{-2}$ s$^{-1}$ for the full irrigation, mild stress, and severe stress treatments, respectively. The most pronounced variations in $E_{leaf}$ among irrigation treatments occurred at 12:00 hours, and the highest $E_{leaf}$ values were recorded at 14:00 hours, with respectively averaged values of 10.61 mmol m$^{-2}$ s$^{-1}$, 6.96 mmol m$^{-2}$ s$^{-1}$ and 5.24 mmol m$^{-2}$ s$^{-1}$ for the full irrigation, mild stress and severe stress treatments. Finally, on average, the tree daily mean

$\Psi_s$ for the fully irrigated treatment was -1.18 MPa, while the mild stress and severe stress treatments showed values of -1.65





MPa and -1.99 MPa, respectively. Similarly, the tree daily averaged values of $g_s$ were 0.32 mmol m$^{-2}$ s$^{-1}$, 0.21 mmol m$^{-2}$ s$^{-1}$, and 0.13 mmol m$^{-2}$ s$^{-1}$ for full irrigation, mild stress, and severe stress treatments, respectively. Additionally, the tree daily $E_{leaf}$ values were 7.74 mmol m$^{-2}$ s$^{-1}$, 5.77 mmol m$^{-2}$ s$^{-1}$, and 4.12 mmol m$^{-2}$ s$^{-1}$ for full irrigation, mild stress, and severe stress treatments, respectively.




**Figure 3: Daily course of stem water potential (Ψs), stomatal conductance (gs) and leaf transpiration (Eleaf) for 29th June and 19th August 2022 in almond trees with three different production systems (open vase (MP), central axis and hedgerow) and irrigation treatments (full irrigation, mild stress, and severe stress).**






### 3.1.2 Sap flow transpiration

The $T_d$-SF showed significant differences among production systems, irrigation treatments, PSxTRT and dates (Table 1). The open vase (MP) transpired significantly higher, with an average of 3.13 mm $d^{-1}$ compared to 2.64 mm $d^{-1}$ for the central axis and 2.46 mm $d^{-1}$ for the hedgerow systems. Notably, in hedgerow, the mild stress treatment showed higher $T_d$-SF values
compared to the full irrigation treatment, although the difference was not statistically significant (Table 2).

Figure 4 shows the daily patterns of $T_h$-SF. The $T_h$-SF patterns exhibited variations based on production system, irrigation treatment, and date. The daily pattern may vary between days due to differences in the main weather forcing parameters (see Fig. 2), as well as an irrigation scheduling error that occurred on June 29th at 12:00 hours, coinciding with the ongoing measurements. The error in the irrigation schedule resulted in significant pattern variations, particularly in the severe stress
treatment. In this treatment, $T_h$-SF exhibited a notable increase at 13:00 hours, reaching its peak at 14:00 and 15:00 hours on June 29th in all production systems. The maximum $T_h$-SF rates recorded in the severe stress treatment on June 29th were 0.14 mm $h^{-1}$, 0.20 mm $h^{-1}$, and 0.23 mm $h^{-1}$ for the open vase (MP), central axis, and hedgerow systems, respectively. Conversely, the maximum $T_h$-SF rates in the severe stress treatment on August 29th were observed between 10:00 and 12:00 hours, with 0.10 mm $h^{-1}$, 0.12 mm $h^{-1}$, and 0.07 mm $h^{-1}$ for the open vase (MP), central axis, and hedgerow systems, respectively.
In the full irrigation treatment, the maximum $T_h$-SF rates varied depending on the day and the production system, occurring between 12:00 and 14:00 hours. In the open vase (MP) system, the highest $T_h$-SF values, averaging 0.45 mm $h^{-1}$, were recorded at 14:00 hours. In the central axis system under full irrigation, the maximum $T_h$-SF occurred at 12:00 hours on June 29th and at 14:00 hours on August 19th, with a $T_h$-SF rate of 0.43 mm $h^{-1}$ for both dates. In the hedgerow system, the full irrigation treatment yielded a maximum $T_h$-SF of 0.37 mm $h^{-1}$ on both days, observed at 12:00 hours on June 29th and at 14:00 hours on
August 19th.

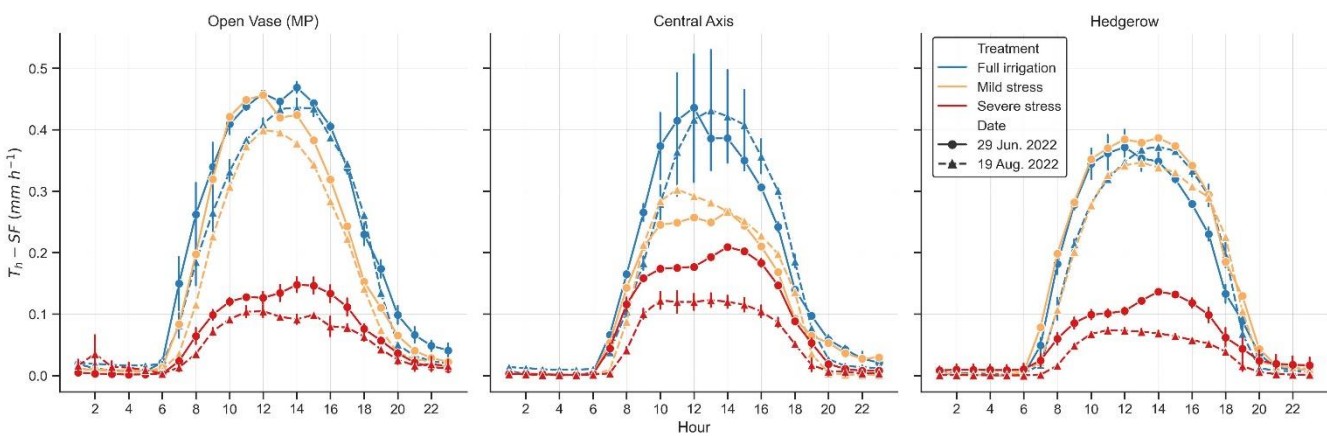



**Figure 4: Daily course of hourly sap flow transpiration ($T_h$-SF) for different irrigation treatments in the production systems a) open vase (MP), b) central axis, and c) hedgerow, for dates 29th June and 19th August 2022.**


Similar to the full irrigation treatment, in the mild stress treatment, the timing of maximum $T_h$-SF depended on the day and the production system. In the mild stress treatment for the open vase (MP), the maximum $T_h$-SF was recorded at 12:00 hours, corresponding to 0.45 mm h$^{-1}$ on June 29th and 0.39 mm h$^{-1}$ on August 19th. In contrast, the mild stress treatment for the central axis system reached its peak at 14:00 hours on June 19th and at 12:00 hours on August 19th, with $T_h$-SF rates of 0.26 mm h$^{-1}$

and 0.30 mm h$^{-1}$, respectively. In the hedgerow system, under the mild stress treatment, the maximum $T_h$-SF rates of approximately 0.38 mm h$^{-1}$ and 0.34 mm h$^{-1}$ were observed at 14:00 hours on June 29th and at 12:00 hours on August 19th, respectively.

The $T_h$-SF exhibited significant differences between 6:00 and 21:00 hours, attributed to the irrigation treatments. $T_h$-SF for the severe stress treatment was systematically lower than the other two treatments. These differences were more evident during

daytime hours. Thus, the maximum differences between the full irrigation and severe stress treatments were observed at 12:00 hours, reflecting an averaged difference of 0.28 mm h$^{-1}$. Furthermore, nocturnal fluxes, which accounted for approximately 5% of the total $T_d$-SF, were observed, with the exception of one tree in the open vase (MP) and one tree in the hedgerow system (both under the severe stress treatment) where nocturnal $T_h$-SF contributed to 21.3% and 10.6% of the total $T_d$-SF, respectively. The statistical analysis showed that $T_h$-SF during the morning (6:00 to 10:00 hours) and afternoon (14:00 to 18:00 hours)

showed significant differences among production systems and PSxTRT (Table 1). During those daytime intervals, the open vase (MP) production system demonstrated significantly higher T compared to the other production systems. The significance of PSxTRT is explained by the fact that the hedgerow, under the mild stress treatment, exhibited higher $T_h$-SF values than the full irrigation treatment in both time periods. Notably, although there was no statistical difference between production systems at midday (11:00 to 13:00 hours), the irrigation treatment was significant for mean $T_d$-SF (Table 1).

Figure 5 illustrates the relationship between the $T_h$-SF measured during the days of the flight campaign and the key parameters utilized in the estimation of $T_d$ ($Rs$, ETo, and ETp) for all irrigation treatments. $T_h$-SF was strongly correlated with $Rs$, ETo and ETp for all irrigation treatments. Overall, the relationship between $T_h$-SF and ETo had the highest Pearson correlation coefficient ($r$), with values of 0.95, 0.95 and 0.90 for the full irrigation, mild stress and severe stress treatments, respectively. Similarly, the correlation with $Rs$ yielded $r$ values of 0.94, 0.94, and 0.87, while ETp showed $r$ values of 0.94, 0.94, and 0.85,

respectively for the full irrigation, mild stress, and severe stress treatments. The ETp model exhibited a root mean squared error (RMSE) of 0.22 mm h$^{-1}$ compared to $T_h$-SF for the full irrigation treatment. Additionally, the RMSE of the ETp model showed significant variation between production systems, with an error of 0.18 mm h$^{-1}$ for the open vase (MP), 0.19 mm h$^{-1}$ for the central axis, and 0.27 mm h$^{-1}$ for the hedgerow systems.

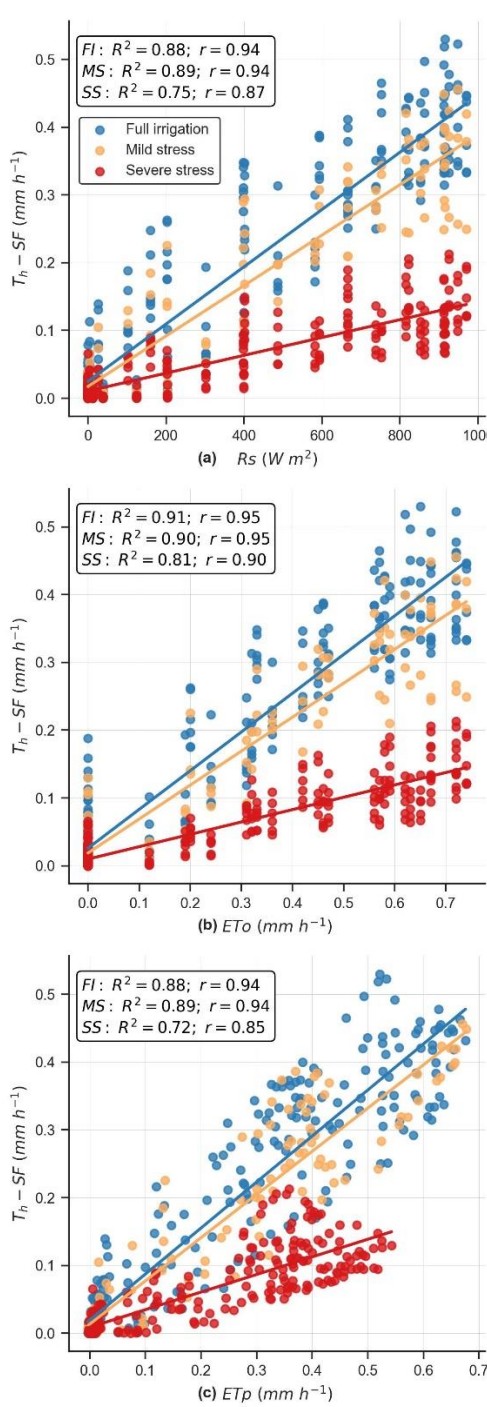


**Figure 5: Regression between hourly sap flow transpiration ($T_h$-SF) with a) solar irradiance ($Rs$), b) reference evapotranspiration (ETo) and c) potential evapotranspiration (ETp), separating by irrigation treatment. The box displays the statistical values for the determination coefficient ($R^2$) and Pearson's correlation coefficient ($r$) across the full irrigation, mild stress, and severe stress treatments.**





The difference between the hourly and daily ratio of $Rs$ ($\%_{RS}$), ETo ($\%_{ETo}$) and ETp ($\%_{ETp}$) and T-SF is shown in Fig 6. The diurnal pattern in $\%_{RS}$ was significantly different between irrigation treatments, but not in production systems. The $\%_{RS}$ displayed a relatively consistent trend between 9:00 and 15:00, fluctuating within the range of 28 to 58% primarily influenced by irrigation treatment and date. However, during the interval from 12:00 to 15:00, the $\%_{RS}$ did not show significant differences

across production systems, irrigation treatments and dates. During this interval of time, the overall average values of $\%_{RS}$ were -14.47, -15.70, -10.2 and -2.47% from 12:00 to 15:00 hours, respectively.





**Figure 6. Daily evolution of difference between hourly and daily mean of αRS, αET$_o$ and αET$_p$. α represents the ratio**
**between transpiration and the reference variable, while '%' corresponds to the formula $(\alpha_{Hour} - \alpha_{Day}) \times \alpha_{Day}$, where**
**the subindex indicates the respective method.**



For its part, $\%_{ETo}$ exhibited a distinct diurnal pattern between the two dates. Values remained relatively constant between 9:00 and 16:00, ranging from -16.44% to 18.06% on both dates. Similar to $\%_{RS}$, $\%_{ETo}$ showed no significant differences between irrigation treatment and date from 12:00 to 14:00 hours. During this time interval, the mean $\%_{ETo}$ values were -4.87% at 12:00, -8.52% at 13:00, and -4.71% at 14:00. The interaction between irrigation treatment and date began to display significant differences from 14:00 to 18:00, with the severe stress treatment on June 29th showing significantly higher values. On the other hand, the $\%_{ETp}$ pattern exhibited significant variations depending on irrigation treatment and date from 7:00 to 10:00 hours, and after 17:00 hours. However, between 11:00 and 16:00 hours, $\%_{ETp}$ did not exhibit any significant effects attributed to production system, irrigation treatment, or date. Between 11:00 and 16:00 hours, $\%_{ETp}$ ranged from -1.16% to 13.07%, with the minimum percentage difference recorded at 11:00 hours (1.16%) on June 29th and 12:00 hours (3.10%) on August 19th.

Figure 7 shows the relative RMSE (RRMSE) and bias (Rbias) when estimating $T_d$ from $T_h$-SF. The $T_d$-SF-$EF_{sim}$ exhibited an RRMSE ranging from 2.7% to 26% after 7:00 hours. Overall, the lowest RRMSE for $T_d$-SF-$EF_{sim}$ was observed at 14:00 hours. However, the RRMSE of $T_d$-SF-$EF_{sim}$ showed significant variability among irrigation treatments at this time, with values of 8.22%, 5.28 % and 17.4 % for the full irrigation, mild stress and severe stress treatments, respectively. Conversely, the RRMSE of $T_d$-SF-$EF_{sim}$ varied as a function of date in the severe stress treatment after 12:00 hours. While on June 29th the RRMSE decreased, on August 19th it increased with values of 26.14% at 14:00 and 25.36% at 16:00 hours, respectively.

The RRMSE of $T_d$-SF-$Rs$ and $T_d$-SF-ETo varied with irrigation treatment. In the full irrigation and mild stress treatments, the RRMSE of $T_d$-SF-$Rs$ showed a convex shape throughout the day, with higher values in the early morning and late afternoon. In the full irrigation treatment, the RRMSE of $T_d$-SF-$Rs$ steadily decreased until 15:00 hours, reaching an average minimum value of 8.4%. In the mild stress treatment, the RRMSE of $T_d$-SF-$Rs$ remained relatively constant at around 10% between 9:00 and 16:00, reaching its lowest point at 15:00 (RRMSE of 4.18%). In the severe stress treatment, the RRMSE of $T_d$-SF-$Rs$ exhibited a sinusoidal curve pattern. In this treatment, the RRMSE of $T_d$-SF-$Rs$ hovered around 15.3% at 9:00 and began to increase until 12:00 or 13:00 hours, with a mean RMSE ranging between 29.83% and 34% depending on the date. On both dates, the RRMSE of Td-SF-$Rs$ decreased at 15:00, with an RMSE of 7.37% and 12.08% on June 29th and August 19th, respectively.



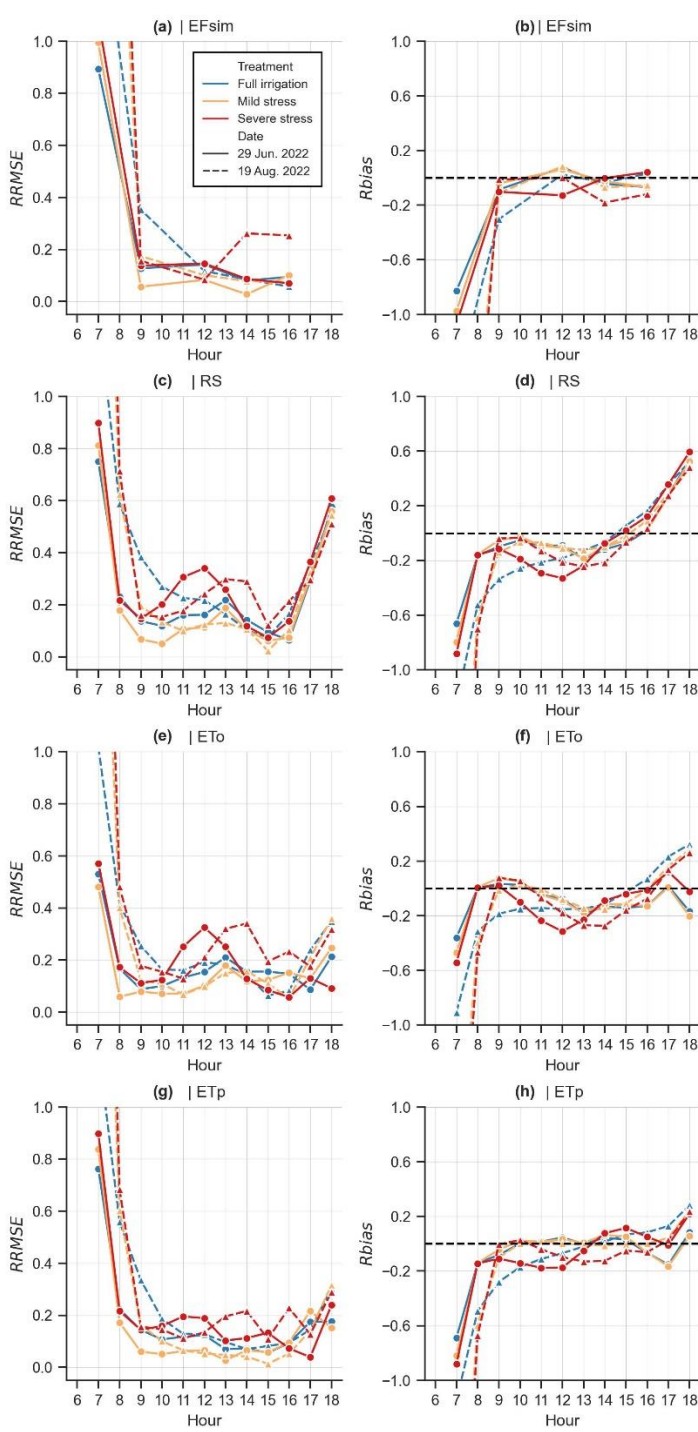

**Figure 7. Relative RMSE (RRMSE) and bias (Rbias) calculated for daily transpiration estimates obtained through the**
435 **EF$_{sim}$, Rs, ETo, and ETp methods using T$_h$-SF at the time when ETo was greater than 0 mm h$^{-1}$.**





The time at which the minimum RRMSE of $T_d$-SF-ETo occurred varied depending on the interaction between irrigation treatment and date. In the full irrigation treatment, the lowest RRMSE of $T_d$-SF-ETo, corresponding to 8.59%, was observed at 17:00 on June 29th. Conversely, in the full irrigation treatment, the minimum RRMSE of $T_d$-SF-ETo (6.29%) was recorded at 15:00 on August 29th. For the mild stress treatment, the minimum RRMSE of $T_d$-SF-ETo on June 19th was 5.86% and was recorded at 8:00, while the minimum RRMSE of $T_d$-SF-ETo on August 29th was observed at 16:00, corresponding to 5.27%. Similar to $T_d$-SF-$Rs$, the RRMSE of $T_d$-SF-ETo in the severe stress treatment presented a sinusoidal curve. The RRMSE of Td-SF-ETo decreased until approximately 10:00 or 11:00 before gradually increasing around noon. On June 29th, it reached a maximum value of 32.50% at 12:00. After 12:00, the RRMSE of $T_d$-SF-ETo began to decrease and reached 5.70% at 16:00 hours. On August 29th, a maximum RRMSE of $T_d$-SF-ETo was observed at 14:00 hours, reaching 34.01%, whereas the minimum RRMSE of $T_d$-SF-ETo was recorded at 11:00 (RRMSE of 12.83%). Finally, the RRMSE of $T_d$-SF-ETp exhibited higher values in the early morning and late afternoon but remained constant from 9:00 to 17:00. Although the severe stress treatment exhibited the highest RMSE of $T_d$-SF-ETp, no significant differences were detected among production systems, irrigation treatments or dates. Overall, the minimum RRMSE of $T_d$-SF-ETp of 7.51% was recorded at 15:00 hours.

### 3.2 Regression of measured and remotely estimated $T_h$ with the TSEB-2T

Figure 8 shows the concurrence between $T_h$-TSEB and $T_h$-SF. The most accurate estimation was obtained at 14:00 hours, with an RRMSE of 29% and an $R^2$ value of 0.81. Comparable error statistics were obtained at 12:00 hours, with an RRMSE of 39% and an $R^2$ value of 0.71. The least favorable outcomes were observed during early morning flights, specifically at 7:00 hours, when the TSEB-2T model provided null estimations for multiple trees. Overall, $T_h$-TSEB showed an overestimation at all hours. However, these overestimations were more pronounced at 9:00 and 16:00 hours, which respectively corresponded to RRMSE values of 77% and 59%.





**Figure 8. Regressions between measured and estimated hourly transpiration with the TSEB-2T model and high-resolution images by hour, production system and irrigation treatment.**





Table 3 presents an ANOVA analysis aimed at assessing the sensitivity of $T_h$-TSEB to production system, irrigation treatment, and date. The estimations of $T_h$-TSEB indicated significant differences among irrigation treatments across all flight times. The three irrigation treatments could be differentiated using the estimations at 9:00, 12:00 and 14:00 hours. On the other hand, the

$T_h$-TSEB at 7:00 hours in the mild stress treatment presented similar values compared with the full irrigation treatment. At 16:00 hours, the mild stress treatment showed comparable $T_h$-TSEB estimations with the full irrigation and severe stress treatments. The production system presented important differences in $T_h$-TSEB when considering the flights conducted at 9:00 and 16:00 hours. At these times, the open vase (MP) production system exhibited significantly higher $T_h$-TSEB. In contrast, the flights conducted at 7:00, 12:00 and 14:00 hours estimated similar $T_h$-TSEB values between production systems.


| Hour | Date | PS | TRT | PSxDate | TRTxDate | PSxTRT | PSxTRTxDate |
|------|------|-----|------|---------|----------|--------|-------------|
| 7:00 | <.0001 | ns | 0.0032 | ns | 0.0282 | ns | ns |
| 9:00 | <.0001 | 0.0062 | <.0001 | ns | ns | ns | ns |
| 12:00 | <.0001 | ns | <.0001 | ns | ns | ns | ns |
| 14:00 | 0.002 | ns | <.0001 | ns | ns | ns | ns |
| 16:00 | ns | 0.012 | 0.0032 | ns | ns | ns | ns |

**Table 3. Analysis of variance (three-way ANOVA) testing the effects of date, production system (PS) and irrigation treatment (TRT) on $T_h$-TSEB for each hour of flight conducted.**

Table 4 shows the influence of production system, irrigation treatment and date on the squared error of $T_h$-TSEB compared

with $T_h$-SF. The results indicate that $T_h$-TSEB using the flight performed at 7:00 hours on June 29th exhibited a systematic error, generating important differences in the squared error due to the date. Furthermore, the squared error showed significant differences between irrigation treatments only for the flights conducted at 9:00. Additionally, a significant effect attributed to the interaction of production system and irrigation treatment was observed at 9:00 hours. Notably, the open vase (MP) under severe stress treatment exhibited a higher error at 9:00 hours. Moreover, while the production system, treatment, and date did

not have a significant impact on the RMSE of $T_h$-TSEB for flights conducted at 12:00, 14:00, and 16:00 hours (Table 4), it is noteworthy that the severe stress treatment consistently exhibited a higher error across all flight hours (Table 5).

| Hour | Date | PS | TRT | PSxDate | TRTxDate | PSxTRT | PSxTRTxDate |
|------|------|-----|------|---------|----------|--------|-------------|
| 7:00 | 0.0004 | ns | ns | ns | ns | ns | ns |
| 9:00 | ns | ns | 0.0386 | ns | ns | 0.0087 | ns |
| 12:00 | ns | ns | ns | ns | ns | ns | ns |
| 14:00 | ns | ns | ns | ns | ns | ns | ns |
| 16:00 | ns | ns | ns | ns | ns | ns | ns |

**Table 4. Analysis of variance (three-way ANOVA) evaluating the effects of date, production system (PS) and irrigation treatment (TRT) on the root mean squared error (RMSE) of $T_h$-TSEB for each hour of flight conducted.**


| Treatment | 7:00 | 9:00 | 12:00 | 14:00 | 16:00 |
|-----------|------|------|-------|-------|-------|
| Full irrigation | 0.059 | 0.142 ab | 0.106 | 0.069 | 0.104 |





| Treatment | 7:00 | 9:00 | 12:00 | 14:00 | 16:00 |
|---|---|---|---|---|---|
| Mild stress | 0.049 | 0.096 b | 0.089 | 0.037 | 0.093 |
| Severe stress | 0.057 | 0.167 a | 0.113 | 0.105 | 0.15 |

**Table 5. Root mean squared error (RMSE) of $T_h$-TSEB (mm h$^{-1}$), categorized by irrigation treatment, for each hour of flight conducted.**

### 3.3 Evaluation of daily upscaling methods to estimate $T_d$ with the TSEB-2T

The $T_d$ was estimated using the different upscaling methodologies and with the 14:00 hours $T_h$-TSEB estimation as starting point (Fig. 9). The $T_h$-TSEB at 14:00 hours was selected due to the highest accuracy obtained (Fig. 8) when validated against the $T_h$-SF. Overall, the results indicate that the $EF_{sim}$, $Rs$ and ETo upscaling methods yielded similar results, even reducing the RRMSE obtained by $T_h$-TSEB. In contrast, the ETp methods exhibited higher RMSE than those obtained by $T_h$-TSEB. The $T_d$-TSEB-$Rs$ and $T_d$-TSEB-ETo reached the highest accuracy, showing an RMSE (RRMSE) of 0.62 mm d$^{-1}$ (23%) and 0.61

mm d$^{-1}$ (22%), respectively. The $T_d$-TSEB-$EF_{sim}$ and $T_d$-TSEB-ETp approaches had RMSE (RRMSE) values of 0.72 mm d$^{-1}$ (26%) and 0.89 mm d$^{-1}$ (32%), respectively. In addition, the $T_d$-TSEB-$EF_{sim}$ and $T_d$-TSEB-ETp yielded larger overestimations, with biases of 0.38 mm d$^{-1}$ and 0.61 mm d$^{-1}$, compared to $T_d$-TSEB-$Rs$ and $T_d$-TSEB-Eto, which had biases of 0.22 mm d$^{-1}$ and 0.14 mm d$^{-1}$.





**Figure 9.** Regressions between measured ($T_d$-SF) and estimated daily transpiration ($T_d$-TSEB) by production system and irrigation treatment with the following upscaling methodologies: a) $EF_{sim}$, b) $Rs$ c) ETo, and d) ETp.

Table 6 shows an ANOVA analysis performed to detect the sensitivity of $T_d$-TSEB-$EF_{sim}$, $T_d$-TSEB-$Rs$, $T_d$-TSEB-ETo and $T_d$-TSEB-ETp to irrigation treatment, production system and date. The results indicate that all approaches exhibited significant



differences in the estimated $T_d$ attributed to irrigation treatment and date. Finally, an ANOVA analysis was conducted to assess the influence of each upscaling method on the RMSE, considering production system, irrigation treatment and date (Table 7). The RMSE in $T_d$-TSEB-$EF_{sim}$, $T_d$-TSEB-$Rs$, $T_d$-TSEB-ETo and $T_d$-TSEB-ETp varied significantly due to irrigation treatment. All the daily upscaling methods resulted in significantly higher RMSE values in the severe stress treatment (Table 8).


| Model | Date | PS | TRT | PSxDate | TRTxDate | PSxTRT | PSxTRTxDate |
|---|---|---|---|---|---|---|---|
| $T_d$-TSEB-$EF_{sim}$ | 0.0002 | ns | <.0001 | ns | ns | ns | ns |
| $T_d$-TSEB-$Rs$ | 0.0007 | ns | <.0001 | ns | ns | ns | ns |
| $T_d$-TSEB-ETo | 0.0003 | ns | <.0001 | ns | ns | ns | ns |
| $T_d$-TSEB-ETp | 0.0252 | ns | <.0001 | ns | ns | ns | ns |

**Table 6. Analysis of variance (three-way ANOVA) for the evaluation of the effects of date, production system (PS) and irrigation treatment (TRT) on $T_d$ estimated with TSEB-2T using flights conducted at 14:00 hours and $EF_{sim}$, $Rs$, ETo and ETp upscaling methods. P values less than 0.05 were considered statistically significant.**

| Model | Date | PS | TRT | PSxDate | TRTxDate | PSxTRT | PSxTRTxDate |
|---|---|---|---|---|---|---|---|
| $T_d$-TSEB-$EF_{sim}$ | ns | ns | 0.0071 | ns | ns | ns | 0.0375 |
| $T_d$-TSEB-$Rs$ | ns | ns | 0.0094 | ns | ns | ns | ns |
| $T_d$-TSEB-ETo | ns | ns | 0.0183 | ns | ns | ns | ns |
| $T_d$-TSEB-ETp | ns | ns | 0.0420 | ns | ns | ns | 0.0416 |

**Table 7. Analysis of variance (three-way ANOVA) evaluating the effects of date, production system (PS) and irrigation treatment (TRT) on the root mean squared error (RMSE) of $T_d$ estimated with TSEB-2T using flights conducted at 14:00 hours and $EF_{sim}$, $Rs$, ETo and ETp upscaling methods. P values less than 0.05 were considered statistically significant**

| Irrigation treatment | $EF_{sim}$ | $Rs$ | ETo | ETp |
|---|---|---|---|---|
| Full irrigation | 0.55 b | 0.43 b | 0.44 b | 0.66 ab |
| Mild stress | 0.32 b | 0.32 b | 0.37 b | 0.42 b |
| Severe stress | 0.97 a | 0.85 a | 0.83 a | 0.92 a |

**Table 8. Root mean squared error (RMSE, mm d$^{-1}$) of the estimated $T_d$ using the TSEB-2T and $EF_{sim}$, $Rs$, ETo and ETp upscaling approaches by irrigation treatment. Different letters mean significant differences at P < 0.05 using Tukey's honest significant difference test considering the irrigation treatments.**

## 4 Discussion

The timing of measurements is crucial for determining the level of water stress and accurately estimating T fluxes. Our observations indicate that the highest differences in $\Psi_s$, $g_s$, $E_{leaf}$, and $T_h$-SF between irrigation treatments were near solar noon (between 11:00 and 14:00 hours), underlining the importance of considering diurnal variations in plant responses to water stress. This period is often when water stress is most pronounced and plant physiological processes are most affected. Accurate measurements during this critical time frame can provide valuable insights into the impact of water stress on plant behavior and T rates. Therefore, our findings reinforce the conclusion that the best moment to determinate water stress is noon or early






afternoon, considering the maximum peaks of T (Gentine et al. 2007; Delogu et al. 2012) and the maximum differences between water status (Bellvert et al. 2014; Anderson et al. 2021; Tian and Schreiner 2021).

In addition, the $T_h$-TSEB estimated using images obtained at 12:00 and 14:00 hours yielded the most accurate results and were able to detect greater differences between irrigation treatments, while the irrigation treatment did not significantly affect the

RMSE (Tables 3 and 4). This is in line with the findings of Anderson et al. (2021), who showed how earlier overpasses often created uniform maps of ET without differentiating between crop water demand. Additionally, Bellvert et al. (2014) showed that the optimal time for capturing high-resolution thermal images to minimize shade effects and monitor leaf water potential and $T_c$ is around solar noon. The higher overestimations in $T_h$-TSEB estimated using images obtained at 7:00, 9:00 and 16:00 hours could be explained by the shadow that covered the thermal images. The $T_h$-TSEB model demonstrates effective

differentiation between irrigation treatments at all hours, particularly around solar noon, where differences between all irrigation treatments are evident. However, it is noteworthy that $T_h$-TSEB fails to exhibit significant differences between production systems around solar noon (Table 3). This poses a challenge when estimating $T_d$ using upscaling methods, as no model detected variations in $T_d$ by production system, as evidenced by $T_d$-SF (Table 6). This limitation is especially critical when estimating $T_d$ from $T_h$-TSEB in canopies with diverse architectural structures.

While the comparison between actual $T_d$-SF and $T_d$-TSEB-EF$_{sim}$, $T_d$-TSEB-$Rs$, $T_d$-SEB-ETo and $T_d$-TSEB-ETp showed similar results (Fig. 9), $T_d$-TSEB-$Rs$, $T_d$-SEB-ETo enhanced the accuracy of $T_d$ estimates, which is reflected in the reduced RMSE values of 0.62 mm d$^{-1}$ and 0.61 mm d$^{-1}$, respectively. It should be noted that both Cammalleri et al. (2014) and Nassar et al. (2021) reported the $R_S$ method as yielding the best results when used as an upscaling parameter to estimate daily ET. However, our results suggest that the superior performance of $Rs$ and ETo to estimate $T_d$ can be attributed to their capacity to

rectify the overestimation observed in $T_h$-TSEB estimates, rather than their inherent alignment with the $T_h$-SF pattern. The underestimation is clarified by the %$_{RS}$ at 14:00 hours, which ranged from -2.47% to -14.47%, while the %$_{ETo}$ ranged from -4.71% to -8.52% (Fig. 6). Related to our results, van Niel et al. (2012) and Cammalleri et al. (2014) observed a systematic underestimation of estimated daily ET values using the $Rs$ approach for a wide range of ecosystems and weather conditions. In this regard, Anderson et al. (1997) proposed a correction factor of 1.1 to compensate for systematic bias, increasing by 10%

the daily ET estimations. Among the advantages of the $Rs$ method, Cammalleri et al. (2014) highlighted its uniform bias around the acquisition time and throughout the season, in contrast to the EF method, ETo, and the top-of-atmosphere radiance (RTOA) as reference variables. Nassar et al. (2021) also found that the $Rs$ method exhibits less sensitivity to seasonal and climate variations compared to the EF approach and use of the net radiation-to-solar radiation ratio (Rn/Rs). It is important to clarify that both the above cited studies evaluated the EF approach with actual AE measurements using eddy covariance towers as

validation data source.

However, our findings indicate that the correction factor proposed by Anderson et al. (1997) of 1.1 for $T_d$ should be determined by water stress, otherwise it may be deemed inadequate. In addition, the timing of the overpass is crucial in determining a correction factor for using the $Rs$ and ETo approaches, particularly for trees experiencing water stress. For instance, although the minimum RRMSE of $T_d$-SF-$Rs$ of 10% might be achievable at 15:00 hours, the RRMSE of $T_d$-SF-$Rs$ could reach 25-30%





around midday in trees under waters stress. Given the variations of the RRMSE when estimating $T_d$-SF-$Rs$ throughout the day and between days, establishing an appropriate correction factor for water stressed trees presents a challenge. These findings complement those of Cammalleri et al. (2014) and Nassar et al. (2021), who concluded that the $Rs$ method is minimally influenced by the timing of the daytime overpass in unstressed vegetation.

The errors associated with the $Rs$ and ETo approaches could be attributable to fluctuations in the daily patterns of $Rs$, ETo,
and the physiological condition of the tree throughout the day. $Rs$ and ETo exhibited an almost perfect concave shape, with their maximum values occurring at 13:00 hours (Fig. 2), which was the local solar noon at our study site. Conversely, in the full irrigation treatment, while stomatal closure was observed at 12:00, both $E_{leaf}$ and $T_h$-SF either remained steady or even increased until 14:00 hours (Fig. 3 and Fig. 4). The maintenance or increase in $E_{leaf}$ and $T_h$-SF during the early afternoon can be attributed to the rise in $T_a$ and the decrease in RH, consequently leading to an increase in VPD during the afternoon (Fig.
2). The variation in patterns between $Rs$, ETo, and $T_h$-SF resulted in lower hourly values of $T_h$-SF/$Rs$ (or $T_h$-SF/ETo) at midday compared to those observed during the early afternoon (15:00-16:00 hours). On the other hand, $T_h$-SF/$Rs$ (or $T_h$-SF/ETo) in the early afternoon exhibited more representative values for estimating $T_d$ in the fully irrigated treatment from $T_h$-SF (Fig. 6). In contrast, the early water stress, as indicated by the $\Psi_s$, resulted in stomatal closure detected at 9:00 in the severe stress treatment. The impact of stomatal closure can be observed in Fig. 4, where the maximum $T_h$-SF was achieved before noon in
the severe stress treatment, specifically between 10:00 and 12:00 hours. Despite the increase in $Rs$ and ETo, the maximum $T_h$-SF in the severe stress treatment remained nearly constant between 10:00 and 16:00 hours on August 19[th]. On June 29[th], $T_h$-SF in the severe stress treatment increased rapidly, starting from 12:00 hours and reaching its maximum value at 14:00 hours, similar to the full irrigation treatment. As a result, the disparities between $T_h$-SF and $Rs$ (or ETo) were most pronounced at midday (Fig. 6), leading to significant potential underestimations when using midday measurements. While $Rs$ and ETo
decreased, $T_h$-SF remained at its maximum until 16:00 hours. Consequently, the relationship between $T_h$-SF and $Rs$ (or $ET_o$) started to become more representative of $T_d$-SF at 15:00 hours on both dates. Therefore, it appears that the optimal time to estimate instantaneous T for daily estimations would fall in the early afternoon, specifically at 15:00 hours, for both the $Rs$ and ETo approaches in all irrigation treatments.

Regarding the ETo method, in line with the findings of Cammalleri et al. (2014), using ETo as a reference variable produced
results similar to those of the $Rs$ method, indicating that it does not represent an improvement. Moreover, the %$_{ETo}$ pattern varied from one date to another (Fig. 6), introducing uncertainty into the potential upscaling adjustments. Differences in the %$_{ETo}$ patterns between days may be attributable to variations in the aerodynamic properties of the canopy between the reference vegetation and the almond canopy. For instance, Colaizzi et al. (2006) obtained good results applying the ETo method in alfalfa and irrigated cotton but poor results for bare soil in drying conditions. It should also be noted that the microclimatic
conditions at the location of the weather station may differ from those of the study site, introducing uncertainty into estimation of the actual ETo of the orchard under study. These two issues may be a possible limitation when using ETo as a reference variable to estimate $T_d$ fluxes (Cammalleri et al. 2014).



The less favorable results obtained of $T_d$-TSEB-$EF_{sim}$ and $T_d$-TSEB-ETp may be more closely linked to the daily $T_h$-SF pattern, hinting at a potential enhancement to address the observed underestimation of $T_d$ when using the $Rs$ and ETo methods. The

$EF_{sim}$ and ETp methods appear to improve the $T_d$ considering the better RRMSE of $T_d$-SF-$EF_{sim}$ and $T_d$-SF-ETp compared to the $T_d$-SF-$Rs$ and $T_d$-SF-ETo. The improvement of the $EF_{sim}$ method aligns with the daily EF curve observed in previous studies, which does not remain constant but instead exhibits an upward concave shape, especially in non-stressed vegetation (Brutsaert 1992; Lhomme and Elguero 1999; Hoedjes et al. 2008; Delogu et al. 2012). Delogu et al. (2012) showed an improvement in the reconstruction of daily ET for various sites and under different climatic conditions, including low water

stress, using the $EF_{sim}$ methodology. This is in line with our findings if we consider that their "low water stress" conditions align with the mild stress treatment. However, it should be noted that the RRMSE of $T_d$-SF-$EF_{sim}$ may increase when using AE estimations during the afternoon for trees under water stress. This is why the RMSE values are significantly higher in $T_d$-TSEB-$EF_{sim}$ when using the TSEB-2T at 14:00 hours (Table 7). The actual EF shape under water stress during the day differs from the $EF_{sim}$ shape, presenting a flatter profile (Lhomme and Elguero 1999; Hoedjes et al. 2008). The larger differences

between actual EF and $EF_{sim}$ during the afternoon could potentially lead to an underestimation of $T_d$ when using the $EF_{sim}$ method. However, $EF_{sim}$ was able to reduce the RRMSE of $T_d$-SF-$EF_{sim}$ at noon in the severe stress treatment by up to 15.35% and 17.61% compared to Td-SF-ETo and Td-SF-Rs, respectively. Additionally, the $T_d$-SF-$EF_{sim}$ exhibited 5% less RRMSE than $T_d$-SF-ETo and $T_d$-SF-$Rs$ in the full irrigation and mild stress treatments. This indicates that the $EF_{sim}$ method might perform well under certain conditions but may have limitations, especially when applied to severely water stressed trees using

afternoon measurements.

The remarkable similarity in patterns between ETp and $T_h$-SF is particularly surprising, considering the concerns raised by Delogu et al. (2012) regarding the applicability of the ETp method under stress conditions. Delogu et al. (2012) suggested that actual ET and ETp might exhibit different daily patterns due to stomatal closure, potentially causing a negative bias when using ETp as a daily upscaling parameter. However, the daily curve of the ratio between $T_h$-SF and hourly ETp (%$_{ETp}$) mitigates

the sinusoidal shape of %$_{RS}$ and %$_{ETo}$, which otherwise increases exponentially from noon to 18:00 hours. Furthermore, the RRMSE of $T_d$-SF-ETp improves compared to the other methodologies and shows less variability among irrigation treatments and hours compared to using $Rs$ and ETo as the adjustment variable. This improvement was particularly evident at 12:00, where $T_d$-SF-ETp showed approximately 5% less RRMSE in the full irrigation and mild stress treatments, while reducing RRMSE at noon by 10.6% and 12.92% in the severe stress treatment compared to $T_d$-SF-ETo and $T_d$-SF-Rs, respectively.

Moreover, the ETp model may hold an advantage due to its incorporation of distinct aerodynamic and radiative properties associated with various canopy architectures, which influence the $T_h$-SF pattern. The variation in RMSE in the estimation of ETp among production systems likely impacted the sensitivity of the ETp model fit to each specific production system. The absence of significant differences in LAI among production systems could affect the accuracy of ETp. Quintanilla-Albornoz et al. (2023) already showed a discrepancy between measured LAI and the fraction of intercepted photosynthetically active

radiation (fIPAR) at the study site, where the hedgerow presented higher LAI values but low fIPAR levels. Considering that fIPAR represents 45% of the absorbed light spectrum (Campbell and Norman 1998), these results reinforce the idea of



improving the shortwave transmittance model for estimating ET fluxes. Indeed, among the complexities, estimating parameters such as LAI, albedo, and the potential single-leaf stomatal resistance ($rst_{min}$) are considered challenging and can pose difficulties in making ETp estimations suitable for operational purposes (Delogu et al. 2012; Gao et al. 2022). However,

enhanced ETp models and the refinement of crucial inputs like LAI and albedo can streamline and enhance ETp estimations, further enhancing its utility as a parameter for $T_d$ estimation in trees with different canopy architecture. For instance, implementing a more intricate model to estimate ETp, like the Shuttleworth and Wallace two-source model (Shuttleworth and Wallace 1985), could enhance the daily upscaling method.

## 5 Conclusion

This study evaluates four methodologies to estimate $T_d$ from instantaneous measurements. The daily upscaling methods were evaluated using sap flow measurements in almond trees under three different production systems and three irrigation treatments. Additionally, this study analyzed the daily pattern of physiological parameters, such as a $\Psi_s$, $g_s$, $E_{leaf}$ and $T_h$-SF, to determine the best moment to estimate both $T_h$ and $T_d$.

The $T_h$-TSEB model effectively distinguished between irrigation treatments especially at 12:00 and 14:00, when differences

between the three irrigation treatments were apparent. However, the $T_h$-TSEB did not show significant differences between the production systems at that time. Therefore, of the evaluated upscaling methods, none of the models could discern the significant differences in $T_d$ estimates across production systems, as observed in $T_d$-SF. In addition, the upscaling methodologies were less accurate in severely stressed trees. Especially when using $Rs$ and ETo as reference variables, the levels of underestimation exhibited significant variations between irrigation treatments and across different hours.

Underestimation was as high as 30% around noon for trees for trees under water stress using the $Rs$ and ETo methods. Therefore, it is advisable to carefully choose an appropriate time schedule. In this context, the $EF_{sim}$ and ETp methods demonstrated more consistent relationships with $T_h$-SF and mitigated the underestimation observed in all irrigation treatments when using the other methods. For instance, both the $ET_{sim}$ and ETp models reduced the RRMSE by 5% in the full irrigation and mild stress treatments using measurements at 12:00 hours. In the severe stress treatment, $EF_{sim}$ reduced the RRMSE by

17.61%, and 15.25% at noon, while ETp reduced it by 10.6% and 12.92% at noon compared to the $Rs$ and ETo methods, respectively.

Moreover, ETp has the advantage of incorporating different aerodynamic and radiative properties associated with production systems. In this sense, the ETp method may be an option to better characterize the $T_d$ in trees with different canopy architectures. In this study, similar LAI estimates between production systems could affect the ETp model, where the hedgerow

system showed a significantly higher error. This situation could impact the sensitivity of the ETp model to differentiate $T_d$ between production systems. One approach to enhance $T_d$ estimations could involve refining the Penman-Monteith ETp model and improving the estimations of parameters such as the LAI, albedo, potential single-leaf stomatal resistance, and the

shortwave transmittance model. Alternatively, using more sophisticated models, such as the Shuttleworth and Wallace two-source model, could also be considered.

**Data availability**

All data mentioned in this document have been generated by the IRTA Water Use Efficiency program team. Data sets produced during this study can be made available upon reasonable request from the corresponding author and/or the Water Use Efficiency program.

**Author contributions**

MQ: conceptualization, data curation, formal analysis, investigation, methodology, software, validation, visualization and writing–original draft preparation. JB: conceptualization, funding acquisition, investigation, project administration, supervision, validation and writing–original draft preparation. HN: Software, validation and writing–review & editing. XM: Funding acquisition and writing – review & editing. AP: Data curation and resources.

**Competing interests**

The contact author has declared that none of the authors has any competing interests.

**Disclaimer**

Publisher's note: Copernicus Publications remains neutral with regard to jurisdictional claims made in the text, published maps, institutional affiliations, or any other geographical representation in this paper. While Copernicus Publications makes every effort to include appropriate place names, the final responsibility lies with the authors.

**Acknowledgements**

The authors would like to thank all the Efficient Use of Water in Agriculture program team, at the IRTA, for their technical support, as well as the Horizon 2020 Research and Innovation Program (H2020) of the European Commission, in the context of the Marie Sklodowska-Curie Research and Innovation Staff Exchange (RISE) action and ACCWA project: grant agreement No.: 823965. To enhance readability and language in this work, the author(s) employed ChatGPT-3.5 throughout the writing
process. Following the use of this tool, the author(s) assumed full responsibility for the publication's content and reviewed and edited it as required.





## Financial support

This research has been supported by the projects ET4DROUGHT (No. PID2021-127345OR-C31) and DIGISPAC (TED2021-
131237B-C21) both funded by the Ministry of Science and Innovation (MICINN-AEI) of Spain.

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
