# Peer review of "Assessment of Upscaling Methodologies for Daily Crop Transpiration using Sap-Flows and Two-Source Energy Balance Models in Almonds under Different Water Status and Production Systems"

_Hydrology and Earth System Sciences, 2024_

## Author Response (AR1)

**RC1 Comments**

**General remaks:**

This paper investigates the temporal upscaling from snapshot (airborne or satellite overpass) to daily transpiration using both sap flow observations and the dual source TSEB model, over an almond orchard. This scaling, which might be very different for woody and herbaceous species, has to my knowledge not been looked at previously, and this paper casts some light on how the stomatal and ecophysiological regulation might affect this temporal extrapolation. The analysis is based on 2 summer UAV flights over a field with contrasted structural (3) and water treatment (3) levels. Main concern is that the outcomes are only based on those 2 dates, therefore on a very limited meteorological forcing conditions. I wonder why in-situ TIR radiometers or cheap TIR imaging cameras have not been used to continuously monitor the 3*3 conditions at representative places to complement the study and offer a longer timeseries.

Thank you for your valuable feedback on our manuscript. We appreciate your thorough review and the insightful comments provided. We have tried to answer all your concerns, and some issues have been added to the main manuscript.

Unfortunately, we lacked the logistics and TIR radiometers required to cover the entire trial. Given that we assessed the TSEB contextual approach (TSEB-2T), it would have been necessary to monitor the canopy temperature and two soil temperatures (on both sides of the row). Consequently, we would have been needed at least 27 sensors to monitor each treatment.

Second concern is that the upscaling using the measured and simulated reference transpiration rates are treated differently just based on a performance, I think it would be useful to see how the performance degradation at other times of the day impact the reconstruction (and not only using 2PM, line 490). If the ref. T is at 10AM for instance, does the TSEB-based extrapolated diurnal transpiration show the same pattern as when using the observed T as a ref ?

The analysis in Figures 6 and 7 already covers the performance degradation resulting from measurement time for each scaling method. Hence, conducting additional analysis on the estimation of daily transpiration's performance degradation due to measurement time would be redundant. Nevertheless, we consider it essential to examine the error linked to the timing of measurements in energy balance models, specifically in the case of TSEB-2T. To address this, we have conducted an estimation of transpiration at five different time points using TSEB. Our primary aim with the overflights is to assess how the timing of flights influences the precision of hourly transpiration estimates, taking into account varying degrees of water stress and diverse canopy architectures, and to investigate its implications for daily scaling methods.

**Minor comments:**

**Line 50**: this is partially true for evapotranspiration ET, not for evaporation E and transpiration T, because the system is often underdetermined (one constraint brought by Land Surface Temperature, whereas there are 2 unknowns, E and T, cf Boulet et al., 2018) except when using TSEB_2T; please comment.

Lines 49-58: We fully agree with your comment, especially when considering TSEB models in general. As you pointed out, the system becomes underdetermined when using radiometric temperature to derive both soil and canopy temperatures. TSEB-2T emerges as a promising option to address this issue by directly deriving energy fluxes separately using both temperatures (soil and canopy), provided that high enough resolution imagery (Nieto., 2019). However, while there are studies evaluating the TSEB-2T for estimating evapotranspiration, the retrieval of transpiration from TSEB-2T has only been validated in a previous work within the same context as this project (Quintanilla et al., 2013).

In summary, I agree with your observation because there is not enough research to indicate the accuracy of transpiration estimation by TSEB-2T in various landscapes. We have reorganized these sentences in the introduction for better coherence. Thank you for your valuable input.

**Line 70**: add both Van Niel et al. (2011) paper on the topic and take their outcomes into consideration (convex/concave up shapes of EF for instance)

Line 73: Thank you, we will consider this work.

**Line 129**: "two laterals" what does that mean? Please explain the technical terms on irrigation and horticultural practices, not everyone is familiar with "open vase" etc.

Line 116: The production system is defined by the combination of the planting distance (distance between trees and rows) in the orchard and the pruning technique employed. For additional clarity, photographs illustrating each of the production systems have been included in Figure 1. Let us know if further clarification is needed.

Line 131: It refers to "two lateral pipes". I have added "two lateral pipes" to the text.

From an agronomic perspective, irrigation systems depend on the spacing between trees and rows. Given that the open vase (MP) system has greater spacing between trees and rows, it is common to install two lateral pipes for irrigation. Conversely, the central axis and the hedgerow systems have narrower spacing between trees and rows, allowing them to be irrigated with just one lateral pipe.

**Line 230 and Line 540:** how did you take into account the shadows for the soil and canopy temperatures ? (cf. Mwangi et al., 2023). What is the impact of the shadows actually ? (needs more analysis and comment than what is written in line 535 to 539)

Shaded pixels were not excluded from the calculation of soil and canopy temperature. The total canopy and soil temperature (all pixels) should be considered to estimate ET fluxes in TSEB-2T models. Therefore, the TSEB model does not account for shadow effects and could be sensitive to Sun position and shadow. We add more analysis to that point between lines 545 and 553.

*The thermal images captured at 7:00, 9:00 and 16:00 hours were more susceptible to thermal radiation directionality (TDR) and shadow effects resulting from the higher zenith angle of the sun. Moreover, the significant contrast between inter-row soil and canopy leads to considerable directional variability in the thermal images (Mwangi et al., 2023). Although TSEB-2T accounts for radiation directionality when estimating H (Norman et al., 1995) and shortwave transmittance (Parry et al., 2019), it may still be susceptible to shadow effects because it does not distinguish between sunlit and shaded sources. To address this issue, Mwangi et al., (2022) proposed a four-component scheme (SPARSE4) as an option to account for sunlit/shaded soil/vegetation energy sources. This scheme couples a dual-source energy balance (SPARSE) model with the physically based unified four-component radiative transfer (UFR97) model.*

**Line 285**: how did you choose the minimum surface resistance for the PM equation ?

Line 288: The minimum surface resistance was estimated using a formulation proposed by Monteith (1995) and Leuning (1995). This formulation takes into account the effect of VPD in stomatal resistance.

*The minimum bulk canopy resistance (rC) for the ETp model was determined through a method that parameterizes the relationship between gs and VPD, as describe by Kustas et al., (2022)*

**Table 1 and 2**: review the column heads which are not self understandable; there is an error in the header of the fourth col. Of Table 2 (twice fc)

Thanks for the comment, the titles were changed for: Table 1. Analysis of variance (three-way ANOVA) testing the effect of Date, production system (PS) and irrigation treatment (TRT) and their interaction on fractional canopy cover (fc), canopy height (hc) and leaf area index (LAI), stem water potential ($\Psi$s), and hourly (Th-SF) and daily transpiration (Td-SF) measured by sap flow sensors. P values less than 0.05 were considered statistically significant.

Table 2: Comparison of the on fractional canopy cover (fc), canopy height (hc) and leaf area index (LAI), and daily transpiration (Td-SF) measured during the flight campaign. Different letters mean significant differences at $p < 0.05$ using Tukey's honest significant difference test considering the interaction between production system and irrigation treatment.

I find the ANOVA analysis not very conclusive and little insightful, maybe worth trying to gain info from a temporal analysis for modelled reference as suggested above, not only 2PM ?

As previously mentioned, we are confident that the analysis presented in figures 6 and 7 adequately addresses the performance degradation associated with measurement time for each scaling method.

Line 599 "hinting at a potential enhancement to address the observed underestimation": this is simply impossible to understand, please rephrase and clarify !

 Lines 611: The sentence was modified.

**RC2 Comments**

The author assesses the upscaling methodologies for daily crop transpiration considering different water stress levels and different production systems. The data presented in this study is very valuable, and will make a great contribution to the community for studying this topic, if the data could be made publicly accessible. Such open science practice will also increase the impact of authors' work.

Thank you for your recommendation. Unfortunately, the data used in this study are still being utilized in other ongoing work within our laboratory. However, the data may be made available upon request, as indicated in the data availability section.

In the Section 2.3.1, soil water potential, stomatal conductance etc. were measured to determine the best moment to estimate both Th and Td. On the other hand, this manuscript does not use soil water potential, stomatal conductance etc. to quantify/estimate daily crop transpiration under different water status? Please authors clarify.

First, there seems to be a misunderstanding. Soil water potential was not measured in this study. Instead, we measured stem water potential, stomatal conductance, and leaf transpiration to evaluate the daily pattern of almond trees under different levels of water stress and their relationship with measurements of actual transpiration.

We believe that it was unnecessary to estimate transpiration using other methods because actual transpiration was obtained through (1) the sap flow sensor and (2) the TSEB models. The use of these two methods is sufficient to achieve the main objective of this study.

In the abstract, the author claimed that 'The improvement of ETp estimations or more sophisticated ETp models could solve this issue'. While the author suggests briefly that canopy architecture/stucture could be a direction to pursue further to improve ETp estimation. It does not explain why this is deemed important, and what is the pathway forward to include this into the estimation.

Thank you for your comments. We have tried to clarify these issues by adding the following sentences to the abstract:

*The use of ETp as a reference variable could address this issue, as it incorporates various aerodynamic and radiative properties associated with different canopy architectures that influence the daily Th-SF pattern. However, more accurate ETp estimates or more advanced ETp models are needed.*

We believe that the discrepancy between leaf area index (LAI) and the fraction of intercepted photosynthetically active radiation (fIPAR) may influence the accuracy of ETp model and, consequently, its use as an upscaling parameter. However, assessing ETp is not an objective of this work. Therefore, a more thorough evaluation of ETp models should be undertaken in future research. A more detailed analysis of this issue can be found in the discussion section, between lines 640 and 655.

This reviewer is wondering why the author only focus on the potential improvement on ETp model, but not on actual ET? If canopy stucture is important for ETp, is it not important for actual ET?

We fully agree with your observation. We emphasized the importance of improving shortwave transmittance models for estimating ET fluxes between lines 645 and 650. We chose not to elaborate further on this point because it was already covered in a previous paper (DOI: 10.1007/s00271-023-00888-1). However, we wanted to highlight the enhancement of ETp models due to their potential as scaling parameters, which is the main objective of this study.

Reference: Quintanilla-Albornoz, M., Miarnau, X., Pelechá, A., Casadesús, J., García-Tejera, O., and Bellvert, J.: Evaluation of transpiration in different almond production systems with twosource energy balance models from UAV thermal and multispectral imagery, Irrig. Sci., https://doi.org/10.1007/s00271-023-00888-1, 2023

**Editor Comments.**
The authors present a relevant study on the temporal upscaling of transpiration data. The two reviewers acknowledge the added value of your study as it provides new insights in the diurnal pattern of transpiration under different water availability regimes. As the study is only done for 2 days, the validity of the study is limited. I recommend the authors to discuss potential limitations of this short validation period.

Furthermore, I would encourage the authors to make the data available according to the HESS-data policy (https://www.hydrology-and-earth-system-sciences.net/policies/data_policy.html). And if this is not yet possible clearly define when and how this will become available

It is certain that as these methodologies are evaluated over more days, the results will become stronger and more reliable. However, the main objective of this study was to assess the response of different daily upscaling patterns in almond trees under varying water status. The initial hypothesis was that trees under water stress (stomatal closure) might not follow the same pattern as net radiation, regardless of weather conditions. Given that significant differences in water status were observed on both measurement dates, we consider the conclusions obtained to be sufficiently robust to address the study hypothesis. Additionally, it is important to note that the meteorological conditions on these two dates were representative of the summer months at the study site. While it would be interesting to evaluate this methodology under different meteorological conditions, such as very high temperatures (e.g., heat waves) or cloud cover, this may be part of a future study.

On the other hand, data sets produced during this study can be made available upon reasonable request from the corresponding author and/or the Efficient Use of Water in Agriculture Program.

---

## Author Response (AR2)

Please add a discussion on the potential limitations of this study due to only a 2-day comparison.

Thank you for your feedback. We have addressed your suggestion by adding the following sentence at the end of the discussion:

"This study was conducted over two measurement days with meteorological forcing conditions representative of typical summer days at the study site (Fig. 2). Additional measurement days would allow for the consideration of wider range of meteorological forcing conditions and vegetative stages in almond trees for a more robust assessment of daily T pattern. However, the data collected effectively represent trees under varying levels of water stress (Fig. 3), consistent with the conditions necessary to address the hypothesis of this work."